# Unraveling Pediatric Group A Streptococcus Meningitis: Lessons from Two Case Reports and a Systematic Review

**DOI:** 10.3390/microorganisms13051100

**Published:** 2025-05-09

**Authors:** Lavinia Di Meglio, Maia De Luca, Laura Cursi, Lorenza Romani, Mara Pisani, Anna Maria Musolino, Stefania Mercadante, Venere Cortazzo, Gianluca Vrenna, Paola Bernaschi, Roberto Bianchi, Laura Lancella

**Affiliations:** 1School of Pediatrics, University of Rome Tor Vergata, 00133 Rome, Italy; 2Infectious Diseases Unit, Bambino Gesù Children’s Hospital, IRCCS, 00165 Rome, Italy; laura.cursi@opbg.net (L.C.); lorenza.romani@opbg.net (L.R.); stefania.mercadante@opbg.net (S.M.); laura.lancella@opbg.net (L.L.); 3Hospital University Pediatrics Clinical Area, Bambino Gesù Children’s Hospital, IRCCS, 00165 Rome, Italy; mara.pisani@opbg.net (M.P.); amcaterina.musolino@opbg.net (A.M.M.); 4Unit of Microbiology and Diagnostic Immunology, Bambino Gesù Children’s Hospital, IRCCS, 00165 Rome, Italy; venere.cortazzo@opbg.net (V.C.); gianluca.vrenna@opbg.net (G.V.); paola.bernaschi@opbg.net (P.B.); 5Anaesthesia, Emergency and Pediatric Intensive Care Unit, Bambino Gesù Children Hospital, IRCCS, 00165 Rome, Italy; roberto.bianchi@opbg.net

**Keywords:** *Streptococcus pyogenes*, meningitis, children, pediatrics, group A streptococcus, GAS

## Abstract

*Streptococcus pyogenes* meningitis is a rare invasive disease, accounting for less than 2% of bacterial meningitis. We presented two case reports and conducted a systematic review using PUBMED, covering the database from its inception up to 31 December 2024, of pediatric cases of Streptococcus pyogenes meningitis. Only case reports and case series were included. Differences in clinical and laboratory parameters were compared between uneventful course and complicated admissions. A total of 57 cases were included. The median age at diagnosis was 4 years. A primary infection focus outside the brain was identified in 61.39% of cases. *S. pyogenes* was identified from cerebrospinal fluid in 66.66% of cases and from blood in 15.79%. Septic shock occurred in 24.56% of cases, and 36.84% had brain anatomical anomalies. All patients received broad-spectrum empiric antibiotics, while protein-synthesis inhibitors were administered in 26.31% of cases. A total of 17% of patients died, and 28.07% experienced sequelae. The identification of *S. pyogenes* from blood and a Phoenix Sepsis Score ≥ 2 were significantly associated with a complicated clinical course. Our findings may offer useful insights for the clinical management of Streptococcus pyogenes meningitis.

## 1. Introduction

Group A streptococcus (GAS), also known as *Streptococcus pyogenes*, is a Gram-positive bacterium and one of the leading human-restricted pathogens affecting children worldwide [1]. This pathogen can cause both non-invasive disease—pharyngitis, scarlet fever, impetigo—as well as life-threatening invasive diseases [1,2,3]. The most common invasive GAS (iGAS) infections are toxic shock syndrome and necrotizing fasciitis [1,2,3]. The main virulence factors of *S. pyogenes* are the M protein and secreted pyogenic exotoxins (SPE). The M protein is the pivotal virulence factor, functioning as an epithelial adhesion factor and inhibiting bacterial phagocytosis. It is encoded by the *emm* gene, with more than 180 M protein variants detected to date. In iGAS infections, an association with *emm 1*, *emm 3*, SPE A, and SPE B has been described [1,2,3,4]. *S. pyogenes* meningitis is a rare invasive disease, accounting for less than 2% of iGAS cases [5]. Since 2022, an increase in iGAS infections has been reported in many European countries, with a subsequent rise in GAS meningitis cases [6]. We present two case reports and a systematic review to:-highlight the key clinical features, common complications, and outcomes of GAS meningitis;-identify factors associated with a complicated course of disease.

## 2. Case Reports

Case 1:

A fully immunized 4 years and 8 months old girl presented to our Pediatric Emergency Department (ED) with fever, left earache within the previous 48 h, and incoercible vomiting over the preceding 24 h. Her past medical history was unremarkable. On physical examination, she was alert and reactive but appeared unwell with meningeal signs. Vital parameters showed normal blood pressure 108/83 mmHg, mild tachycardia 105 bpm, and mild tachypnea 40/min. Otoscopy showed left acute otitis media (AOM), while the rest of the examination was normal. Therapy with ceftriaxone and acyclovir was initiated, and blood tests, a head CT, and a lumbar puncture were performed (Table 1). Blood tests revealed neutrophilic leukocytosis and elevated C-reactive protein (CRP). Cerebrospinal fluid (CSF) analysis was compatible with bacterial meningitis (Table 1), and the head CT revealed left mastoiditis with a hypodense fluid collection (maximum thickness: 5 mm), involving the temporal and occipital lobes. Blood and CSF cultures were collected, and a rapid multiplex PCR panel for viruses and bacteria on CSF was negative. Seven hours later, the patient developed focal seizures and worsening neurological status. On examination, she was unresponsive, presenting with anisocoria, right fixed mydriasis, decerebrate posturing, and a positive Babinski sign. The patient was intubated, and an emergent head CT revealed cerebral edema. Dexamethasone and mannitol were administered, and an urgent decompressive craniotomy was performed. Vital parameters showed hypotension (80/50 mmHg), tachycardia 170, and tachypnea 40/min. The CSF culture became positive for *Streptococcus pyogenes* after 10 h, and therapy was adjusted with the addition of linezolid, while acyclovir was discontinued. Despite those interventions, the patient developed hypotensive shock, and an electroencephalogram (EEG) showed the absence of electrical activity. The patient was declared brain dead 48 h after admission.

Case 2:

A fully immunized 5 year and 9 months old girl presented to a peripheral ED with a 48 h history of fever and lethargy. On examination, she exhibited meningeal signs. Therapy with ceftriaxone and vancomycin was initiated, and the girl was transferred to our pediatric intensive care unit. Upon arrival, she was lethargic with meningeal signs, but no evidence of localized infection was detected. Vital parameters showed normal blood pressure 105/70 mmHg, tachycardia 112 bpm, and mild tachypnea 42/min. Blood tests revealed neutrophilic leukocytosis and elevated CRP. Blood and CSF cultures were collected. CSF analysis was consistent with bacterial meningitis (Table 1). An initial head CT showed a fluid collection (maximum thickness: 4 mm) in the right frontal and parietal regions. BIOFIRE^®^ FILMARRAY^®^ Meningitis/Encephalitis Panel for viruses and bacteria on CSF was negative. After consultation with microbiologists, given the strong suspicion of bacterial meningitis, the Biofire Blood Culture Identification (BCID) Panel designed for blood—which includes a larger number of bacterial targets—was performed on the CSF, yielding a positive result for *Streptococcus pyogenes*. Therapy with dexamethasone was initiated and therapy with immunoglobulin for 3 days at 400 mg/kg was implemented, while vancomycin was discontinued. After five days, the patient was afebrile but continued to exhibit meningeal signs and developed left hemiparesis; BP was stable 98/75 mmHg, and at lower limit the heart rate was 60 bpm. An EEG revealed asymmetrical brain activity, and a brain MRI showed an enlargement of the fluid collection, now involving the entire right hemisphere with a thickness of 6 mm. Antibiotic treatment was remodulated with addition of linezolid, and an urgent decompressive craniotomy was performed. From that moment, the patient showed a slow but progressive improvement. Physiotherapy was started, and after 10 days, the neurological examination was normal. Audiometry performed prior to discharge was also normal. The patient was discharged home after 21 days in good clinical condition, continuing a tapering of steroid therapy at home. Two years after the event, she remains in good health, with no neurological sequelae.

## 3. Materials and Methods

A systematic review following the PRISMA 2020 statement was conducted from the inception of the database up to 31 December 2024, using PUBMED. The following keywords were used: Streptococcus pyogenes meningitis, Group A Streptococcus meningitis, and invasive Streptococcus pyogenes disease. The inclusion criteria were as follows: a diagnosis of bacterial meningitis based on CSF features and clinical conditions, along with culture or PCR (from either CSF or blood) positive for *S. pyogenes*, and patients aged < 18 years. Only case reports and case series were included. Cases from all selected studies were evaluated and analyzed. Study selection followed the PRISMA flow diagram (Figure 1). Two researchers independently assessed the eligibility of the studies. Data extraction from individual studies was performed in duplicate. Any discrepancies in data extraction were discussed and resolved among the authors.

A total of 306 manuscripts were initially identified. After reviewing titles and abstracts and eliminating duplicates, 238 full-text articles were assessed. Ultimately, 39 papers were selected and included in the review. We analyzed each of the retained studies, as well as our two case reports, to create our own database. The following variables were collected: age, sex, medical history (both past and present), prior *S. pyogenes* localized infections, laboratory results, CT/MRI findings, treatment regimens, and clinical outcomes. In each study, all categories and variables were analyzed. Data were entered into a dedicated database. Categorical data are presented as frequencies and percentages, while continuous data are expressed as means/medians with ranges, depending on the statistical distribution.

Patients were classified into two groups: those with a uneventful meningitis course, defined as patients who did not experience septic shock, neurological deterioration, or brain involvement on CT/MRI, and those with a complicated course, defined as patients who developed septic shock, neurological instability, brain involvement on CT/MRI, sequelae (including nerve palsy, deafness, neurological impairment), or death. The following variables were compared between the two groups: age, female sex, focus of infection outside the brain, CSF white blood cell (WBC) count, CSF protein levels, CSF glucose levels, WBC blood count, identification of *S. pyogenes* on blood, Phoenix Sepsis Score ≥ 2 [7], use of antibiotics with activity on protein synthesis inhibition, dexamethasone therapy, and length of hospital stay. A univariate analysis was initially performed to assess the associations between each independent variable and the outcome. The Chi-square (χ^2^) test was used for nominal variables, with a significance threshold set at 0.05. For independent numerical variables following a normal distribution, comparisons between the two groups were made using a one-tailed *t*-test, with a significance level of 0.05. Subsequently, a multivariate analysis using logistic regression was conducted, adjusted for age and sex, accounting for potential confounding factors. Odds ratios (OR) were calculated, along with 95% confidence intervals (CI), to assess the strength of the associations. All statistical analyses were performed using Jamovi version 2.6.44.

## 4. Results

Here, we report the results of a systematic review of the literature and the description of two cases of *S. pyogenes* meningitis. In total, 39 papers reporting a total of 55 cases were included in the analysis, and we also added 2 cases admitted to our hospital for a total of 57 cases (Figure 1) [8,9,10,11,12,13,14,15,16,17,18,19,20,21,22,23,24,25,26,27,28,29,30,31,32,33,34,35,36,37,38,39,40,41,42,43,44,45,46]. Results are summarized in Table 2 and Figure 2.

The median age at diagnosis was 4 years (range: 1 day–15 years); 54.38% (31/57) were female. Seven percent (4/57) of the cases reported previous predisposing conditions for bacterial meningitis, such as medical devices, HIV infection, and prematurity [10,15,17,28]. A primary focus of infection outside the brain was identified in 61.39% (35/57) of cases, with otitis and skin infections being the most common. Six patients (10.52%) had received a course of *S. pyogenes* sensitive antibiotics for at least 48 h before admission, including azithromycin, amoxicillin, and cephalexin [14,17,32,37,38,40]. All patients presented with fever and meningeal signs, including altered mental status. Most of them also had vomiting, headache, and seizures. The diagnosis was confirmed by CSF features (turbid CSF, low glucose, high protein, high white blood cells) in all cases. GAS was identified from CSF in 66.66% (38/57) of cases, and in three of those cases, only by PCR. In 15.79% (9/57), *S. pyogenes* was isolated from blood, while in 17.55% (10/57), *S. pyogenes* was detected from both CSF and blood. In nine cases, *S. pyogenes* was detected in other tissues, including ear cultures in patients with otitis, skin vesicles in patients with skin infections, and tonsillar exudates in patients with tonsillitis. In one case, *S. pyogenes* was detected on the cervix of the newborn’s mother [45].

Forty-seven percent (27/57) of patients were admitted to the pediatric intensive care unit (PICU). Forty-five percent (26/57) had a Phenix sepsis score > 2. The clinical course was uneventful in 38.59% (22/57). However, 24.56% (14/57) experienced septic shock requiring inotropic support, 33.33% (19/57) entered a coma, 21.05% (12/57) experienced status epilepticus, 8.7% (5/57) had cranial nerve impairments, and 36.84% (24/57) had brain anatomical anomalies detected by cerebral imaging.

A cerebral CT was performed during admission or later in 56.14% (32/57) of cases, with negative results in 25% (8/32) of patients. Brain involvement was detected in 75% (24/32), including 12 cases with an intracranial fluid collection (extra-axial collection *n* = 8, intra-axial collection *n* = 4), 7 cases with vascular involvement, 12 cases of brain edema, 5 cases of brain tissue necrosis, 3 cases of ventriculomegaly, 4 cases of cerebritis/ventriculitis, and 1 case of myelitis [9,12,13,14,16,17,18,19,20,23,24,25,26,27,30,32,35,36,37,39,42,46]. All patients received a broad-spectrum empiric antibiotic therapy with third-generation cephalosporins alone in 66.66% (38/57) of cases. Seventeen percent (10/57) received third-generation cephalosporins plus a glycopeptide, mainly vancomycin. The remaining cases received different antibiotic regimens, including meropenem alone or in combination with linezolid, third-generation cephalosporins plus clindamycin, or cephalosporins/penicillin combined with rifampicin or linezolid. In 50.87% (29/57) of cases, therapy was targeted after GAS identification. In 31.67% (18/57) of cases, therapy was de-escalated to intravenous penicillin based on susceptibility testing. Other regimens included the addition of clindamycin, rifampicin, or linezolid to cephalosporins/penicillin or the substitution of the previous therapy with meropenem. Overall, protein-synthesis inhibitors (clindamycin, linezolid, or rifampicin) were used in 26.31% (15/57) of cases. The median length of therapy was 18 days (range: 10–42 days, *n* = 36). Dexamethasone with or without mannitol was used in 33.33% (19/57) of cases [9,12,17,23,25,27,30,31,32,33,35,36,39]. Other treatments included anticonvulsant therapy for patients with seizures, heparin for those with thrombosis, and supportive care for patients in critical conditions. Immunoglobulins were used in three patients (3.50%) [30,36].

Surgical procedures were performed in 13 patients (22.80%). In particular, three patients had a liquor derivation, two underwent myringotomy/tympanostomy, two had mastoidectomy, two required abscess drainage, two had craniectomy, one had both mastectomy and abscess drainage, and one patient had a combination of mastoidectomy, myringotomy, and craniectomy [9,11,12,16,17,27,32,35,37,42,46].

Seventeen percent (10/57) of patients died, all within the first 48 h of hospitalization, while 28.07% (16/57) had sequelae, including hearing loss, nerve palsy, and varying degrees of neurological impairment defined according to Pediatric Cerebral Performance Category [9,10,11,12,13,14,16,17,19,20,27,28,33,34,35,36,37,39,41,42,43,46]. Sixty-two percent (31/57) had a complete recovery. Table 3 shows the univariate analysis of the comparisons between patients with an uneventful course versus those with a complicated course. Age, sex, presence of predisposing conditions, the primary non-invasive *S. pyogenes* focus, use of dexamethasone, and antibiotics that interfere with protein synthesis were not statistically significantly different between the comparison groups in the statistical analysis. The presence of *S. pyogenes* on blood and a Phoenix Sepsis Score ≥ 2 were significantly associated with a complicated course. Regarding laboratory exams, patients with a complicated course had significantly higher levels of WBC and proteins in the CSF. CSF glucose and blood WBC were not statistically different between the groups. Table 4 reports the logistic regression analysis conducted to identify factors associated with a complicated course adjusted for age and sex: the presence of *S. pyogenes* on blood was significantly associated with a complicated course (OR: 6.101, *p* = 0.01), indicating a more than 6-fold increase in the likelihood of a complicated course in patients with GAS bacteremia. Similarly, a Phoenix Sepsis Score ≥ 2 was strongly associated with a complicated course (OR: 68.570, *p* < 0.001), suggesting that this subset of patients have an higher risk of sequelae and/or death.

## 5. Discussion

Bacterial meningitis is a life-threatening condition that causes high morbidity and mortality in children. The most common pathogens involved are *Streptococcus pneumoniae*, *Neisseria meningitidis*, and *Haemophilus influenzae* type B [47]. *GAS* is responsible for fewer than 2% of bacterial meningitis [5].

Since 2022, the WHO has reported an increase in iGAS infections in at least five European countries, particularly in the age group from 0 to 5 years old [6,48,49,50,51,52,53]. In our tertiary pediatric hospital, between April 2023 and July 2024, we recorded 2 cases of GAS meningitis out of 34 cases of iGAS infections.

*S. pyogenes* meningitis exhibits peculiar features. According to our study and previous reports, in half of the patients, the primary focus of infection was not the brain, with otitis and skin infections being the most common sources of bacterial spread. Meningitis may occur through brain invasion via the hematogenous route or by direct spread through continuity. The hematogenous route could be responsible for dissemination from the skin and the nasopharyngeal mucosa, while direct spread seems to be involved in GAS infections of the ear and mastoid bone [5,9,16,23,54].

The main *S. pyogenes* virulence factors involved in adhesion and bacterial dissemination are the M protein, Streptolysin O, streptococcal DNases, and the IL-8 protease SpyCEP. So far, studies have failed to demonstrate differential expression of these factors in patients with GAS meningitis compared with those with otitis or oropharyngeal colonization. Therefore, host characteristics may represent the key discriminant in this process [55].

To better define the pathogenesis and risk factors for local infection spreading, more studies are advocated [54,55].

Based on our findings and those of previous studies, we recommend some specific diagnostic and therapeutic steps in cases of *S. pyogenes* meningitis.

The initial approach is the same for all bacterial meningitis. In case of suspicion of meningitis, a thorough examination is mandatory, especially to recognize meningeal signs and any source of peripheral infection [47,56,57,58]. It is crucial to collect blood samples and cultures, as well as perform lumbar puncture for the CSF analysis and culture. Broad-spectrum antibiotics should be initiated as soon as cultures are collected; however, the execution of head CT/MRI, blood tests, and lumbar puncture should not delay the administration of antibiotics [47,56,57,58]. If possible, we recommend using molecular techniques, especially in patients who received antibiotics before blood and CSF collection [59]. The most common rapid multiplex PCR panel does not include GAS, but in cases of high suspicion—such as concomitant tonsillitis, otitis, or skin impetigo—a more comprehensive panel should be considered [16]. At our center, when rapid multiplex PCR results on CSF are negative and bacterial meningitis is highly suspected, a more comprehensive panel designed for blood testing on cerebrospinal fluid is performed. This approach has also been suggested in other studies [60,61].

Empiric therapy typically involves ceftriaxone plus vancomycin, depending on local resistance patterns [47,56,57,58]. Once *S. pyogenes* is isolated, the approach should be personalized for each case and involve a multidisciplinary team, including pediatricians, intensivists, neurosurgeons, and infectious disease specialists. The antibiotic regimen should be targeted based on the antibiogram. Penicillin G is usually the antibiotic of choice [56]. *S. pyogenes* meningitis has been associated with specific M types and exotoxin production, particularly SPEA, SPEB, and SPEC. Based on these virulence features, there is a growing trend of using protein-synthesis inhibitors (such as linezolid, rifampicin, and clindamycin) in combination with penicillin or cephalosporins [4,16,20,35,62,63]. In both our patients, we decided to add linezolid, an oxazolidinone that inhibits bacterial protein synthesis by binding to the bacterial 23S ribosomial RNA of the 50S subunit, based on its good cerebral penetration, more favorable pharmacokinetic properties compared to vancomycin, and a theoretical activity against the production of pyrogenic exotoxins A and B [64]. In our study, 26% of patients were treated with this approach. Clindamycin has been reported to have poor CSF penetration, and therefore its role in the management of CNS infections is debated [65,66]. Nowadays, no comparative studies have been conducted comparing beta-lactam monotherapy with beta-lactam plus protein-synthesis inhibitors in *S. pyogenes* meningitis, and further research is needed to define the most effective strategy [67,68].

Dexamethasone may play a positive role in reducing brain edema and the inflammatory cascade. The results of its use in Streptococcus pneumoniae meningitis encourage its administration in *S. pyogenes* meningitis as well. In a 2015 Cochrane meta-analysis and systematic review regarding corticosteroid use for acute bacterial meningitis, the use of steroids in Streptococcus pneumoniae meningitis was proven to be protective against death but it had no significant beneficial effect on hearing loss sequelae [69]. In our analysis, steroid use was not statistically different between patients with an uneventful course and those with a complicated course [35,70,71,72]. Of note, our results cannot define the efficacy of steroids in this setting, primary because of the small sample size and secondly because in many cases, steroids were introduced only after the evidence of brain edema at the MRI, rather than immediately after the detection of the bacteria. Additionally, the efficacy demonstrated in Streptococcus pneumoniae is not enough to advise its use in *S. pyogenes* meningitis. Even if they belong to the same genus (Streptococcus), they are distinct species with different pathogenic profiles. Based on the current literature, GAS meningitis exhibits a higher rate of brain anomalies on brain CT/MRI (17% vs. 36.84%), death (13% vs. 17.54%), and sequelae (23% vs. 28.07%) compared to *S. pneumoniae* meningitis [41,42]. Our results align with previous studies [5,9,29,35,38,72,73,74].

We recommend continuous monitoring of the patient during the first 48–72 h, particularly for those with elevated WBC or protein levels in the CSF, as well as those with GAS bacteremia or signs of sepsis, as these factors have been associated with a complicated course, sequelae, or death in our study. Fatal complications occurred in 17% of our cohort, all of them within the first 48 h; thus an intensive setting should be considered at the time of admission. In the event of worsening of the clinical condition or lack of improvement, we recommend performing additional tests based on the clinical presentation, including imaging, as cerebral parenchyma involvement or vascular alterations are common in *S. pyogenes* meningitis [4,16,20,35,45,73,74]. Surgical intervention should be considered, especially in cases of brain anomalies (e.g., hydrocephalus or fluid collections) or when otitis or mastoiditis is the source of infection. In cases of thrombosis, the use of heparin and high doses of steroids has been proposed [12,37].

Looking to the future, some considerations need to be done in terms of possible concerns in the management of *S. pyogenes* meningitis. In the last three years, studies in the Netherlands have reported a concerning increase not only in iGAS infections but also specifically in GAS meningitis. Moreover, while molecular studies before 2022 showed that approximately 35% of isolated strains belonged to the emm1 subtype, after 2022, nearly 90% of cases were associated with the emm1 subtype, and of these, 80% were identified as the toxigenic M1uk variant, known to be associated with more severe clinical courses [75].

Interestingly, the current concern is not limited to the emergence of *S. pyogenes* strains with additional virulence factors but also includes the rise in antibiotic resistance. Penicillin remains the antibiotic of choice; however, over the last 20 years, there has been a troubling trend of reduced penicillin susceptibility, not yet a case of resistance [76]. Several mechanisms have been proposed for this phenomenon: (1) intracellular persistence of *S. pyogenes* within tonsillar tissues, where poor antibiotic penetration reduces treatment efficacy, and (2) beta-lactamase-mediated inactivation, where co-infecting bacteria produce extracellular beta-lactamase that degrades penicillin in the absence of an inhibitor. One of the latest discoveries as mechanisms of resistance involves mutations in the *pbp2x* gene, which encodes a key enzyme in peptidoglycan synthesis [76,77]. These mutations have been linked to reduced susceptibility to penicillins and cephalosporins and are associated with multiple emm types exhibiting high mortality rates. Even more concerning is the growing resistance to macrolides. Over the past decade, the incidence of macrolide resistance has increased dramatically, reaching rates as high as 20–40%. The mechanisms behind this resistance typically involves the methylation of the 23S rRNA target by *erm* (erythromycin ribosomal methylase) genes (ermA and ermB) [76,77,78].

Finally, despite global advances, a reliable vaccination is still not yet available, but a number of vaccine candidates are in early human trials [79].

The limitations of our study include, primarily, the small sample size, mainly due to the rarity of the disease. Another significant limitation is that the data were derived from retrospective studies, each focusing on different aspects of the disease. As a result, some data may have been omitted by the original authors based on the focus they chose for their paper. Additionally, it is important to consider that severe cases may be overrepresented, as they might have been more likely to be published compared to milder cases, introducing a significant bias. Furthermore, the length of follow-up varied across case reports which could have influenced the reporting of sequelae.

## 6. Conclusions

Our study reviewed the management of *S. pyogenes* meningitis. No guidelines or unique approaches to this invasive infection are available yet. Our study may help clinicians quickly identify high-risk patients with *S. pyogenes* meningitis who require more intensive monitoring during hospitalization to reduce morbidity and mortality associated with this infection. Further studies are needed to enhance knowledge for the management of iGAS infections.

## Figures and Tables

**Figure 1 microorganisms-13-01100-f001:**
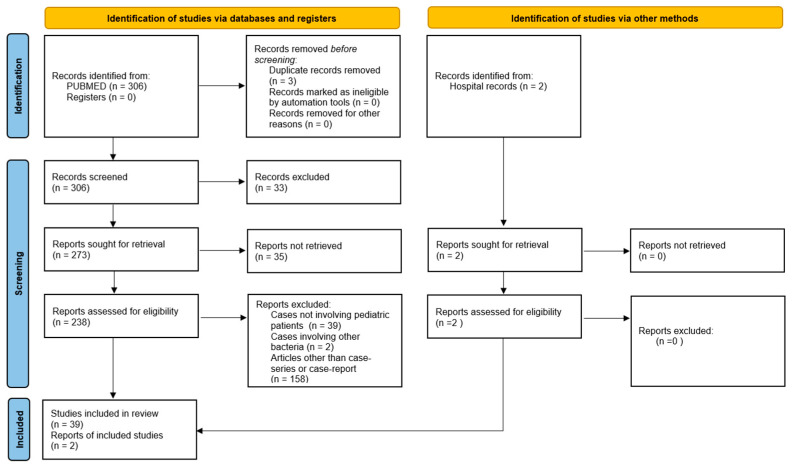
Prima flowchart.

**Figure 2 microorganisms-13-01100-f002:**
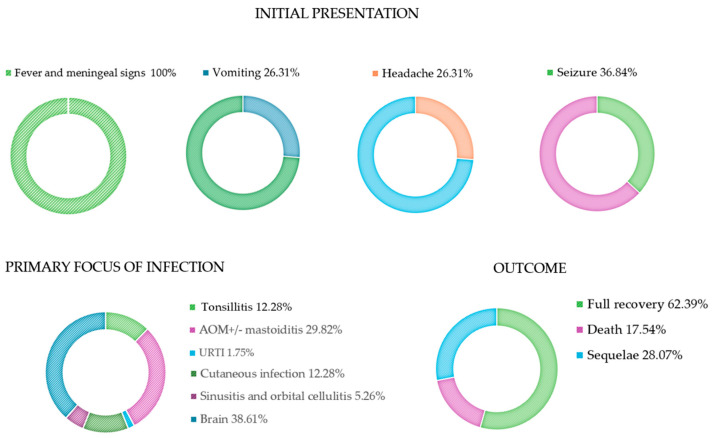
Initial presentation, primary focus of infection and clinical outcome. AOM: acute otitis media; URTI: upper respiratory tract infections.

**Table 1 microorganisms-13-01100-t001:** Clinical and laboratory features of the two case reports.

	Case One	Case Two
Age Sex	4 years and 8 months Female	5 years and 9 months Female
Comorbidities	No	No
Local infection	AOM	No
Blood exams		
White blood cells	32,280/m^3^	18,370/m^3^
Neutrophils	95.54%	94%
Lymphocytes	1.1%	3.0%
Hemoglobin	11.5 g/dL	9.7 g/dL
Platelets	450,000/m^3^	137,000/m^3^
CRP	16.76 mg/dL	26.74 mg/dL
Procalcitonin	28.6 ng/mL	24.3 ng/mL
Glucose	95 mg/dL	140 mg/dL
Cerebrospinal fluid		
Appearance	turbid	turbid
Color	xanthochrome	xanthochrome
White blood cells	296/m^3^	49/m^3^
Glucose	2 mg/dL	58 mg/dL
Proteins	511 mg/dL	125 mg/dL
Treatment and Outcome		
Therapy	Ceftriaxone, Linezolid	Ceftriaxone, Vancomycin, Linezolid, Immunoglobulins
Surgery	Craniectomy	Craniectomy
Length of stay	2 days	21 days
Outcome	Death	Full recovery

**Table 2 microorganisms-13-01100-t002:** Characteristics of the two patients admitted to our hospital and of the patients derived from the systematic review.

Variable	Variable Subclassification	Results	Variable	Variable Subclassification	Results
Predisposing conditions	HIV VPS Cochlear implant Prematurity	1.75% (1/57) 1.75% (1/57) 1.75% (1/57) 1.75% (1/57)	Cerebral TC/RMN findings	Cerebral collection Hydrocephalus Cerebral edema Vasculopathy Necrosis Ventriculitis Myelitis	21.04% (12/57) 5.26% (3/57) 8.77% (12/57) 21.04% (12/57) 8.77% (5/57) 7.01% (4/57) 1.75% (1/57)
Empiric antibiotic therapy	Cephalosporin Cephalosporin plus glycopeptides Other regimes Overall use of protein-synthesis inhibitors	66.66% (38/57) 17.54% (10/57) 15.78 (9/57) 26.31% (15/57)	Additional treatment/ therapies	Steroids Immunoglobulins Surgery	33.33% (19/57) 5.26% (3/57) 22.80% (13/57)
Overall complications	Status epilepticus Nerve involvement Hydrocephalus Brain collection Vasculopathy Hygroma Cerebral edema Coma Phoenix Sepsis Score ≥ 2	21.04% (12/57) 7.01% (4/57) 5.26% (3/57) 21.04% (12/57) 12.28% (7/57) 7.01% (4/57) 24.56% (14/57) 33.33% (19/57) 45.61% (26/57)	Outcome	Full recovery Death Sequelae: -NSHL -Third nerve palsy -Moderate disability -Severe disability	62.39% (31/57) 17.54% (10/57) 28.07% (16/57) 5.26% (3/57) 1.75% (1/57) 5.26% (3/57) 15.80% (9/57)

VPS: ventricular peritoneal shunt, NSHL: sensorineural hearing loss.

**Table 3 microorganisms-13-01100-t003:** Comparison between patients with uneventful and complicated course of disease.

Variable	Uneventful Course N = 22	Complicated Course N = 35	*p*-Value
Age (y)	5.5	5.21	0.41
Age below 1 y	4/22	11/35	0.26
Female sex	12/22	19/35	0.98
Predisposing conditions	2/22	2/35	0.62
Other focus of infection beside the brain	13/22	22/35	0.77
CSF WBC (mg/dL)	1350.3	2586	0.03
CSF proteins (mg/dL)	126.26	252.42	0.02
CSF glucose (mg/dL)	29.25	20.33	0.12
WBC on blood (mmc)	18,836.36	20,216.4	0.35
*S. pyogenes* on blood	3/22	18/35	<0.01
Phoenix Sepsis Score ≥ 2	1/22	25/35	<0.01
Use of protein-synthesis inhibitors	3/22	12/35	0.064
Dexamethasone therapy	4/22	15/35	0.054
Length of stay (d)	17	24	0.02

WBC: white blood cells: CSF: cerebrospinal fluid.

**Table 4 microorganisms-13-01100-t004:** Logistic regression between patients with uneventful and complicated course of disease adjusted for age and sex.

Variable	*p*-Value	Odds Ratio	CI 95%
Predisposing conditions	0.618	0.595	0.0771–4.59
Other focus of infection beside the brain	0.712	1.247	0.386–4.02
*S. pyogenes* on blood	0.01	6.101	1.512–24.62
Phoenix Sepsis Score ≥ 2	<0.001	68.570	7.3733–637.68
Use of protein-synthesis inhibitors	0.06	3.846	0.915–16.16
Dexamethasone therapy	0.05	3.763	0.999–14.18

## Data Availability

The original contributions presented in this study are included in the article. Further inquiries can be directed to the corresponding author. The raw data supporting the conclusions of this article will be made available by the authors on request.

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
