# Peer review of "Unraveling Pediatric Group A Streptococcus Meningitis: Lessons from Two Case Reports and a Systematic Review"

_microorganisms, 2025, doi:10.3390/microorganisms13051100_

Round 1
Reviewer 1 Report
Comments and Suggestions for Authors
Q1:Abbreviations: Define iGAS in the list (currently missing).
Q2: abstract: Only case report and case series were included." → Should be "Only case reports and case series were included."
Q3: abstract: "The identification of GAS from blood and Phoenix Sepsis Score ≥2 were statistically significantly associated..." → Should be "The identification of GAS from blood and a Phoenix Sepsis Score ≥2 were significantly associated..."
Q4: Case Reports: It will be good if the authors can describe more about signs of increased ICP, vital parameters, etc.
Q5: Case Reports: "4-year-8-month-old" → Hyphenation error; use "4 years and 8 months old.
Q6: Case 2: The decision-making process regarding therapy changes should be clarified. Why was vancomycin discontinued but linezolid added?
Q7: Materials and Methods: Please explain what statistical software used for analysis. Also, please clarify whether logistic regression was considered for multivariable analysis.
Q8: Materials and Methods: please clarify the choice of a one-tailed t-test (rather than two-tailed) for comparing continuous variables.
Q9: Please address if the review protocol was registered in a database like PROSPERO.
Q10: In the discussion, since only case reports and case series were included, you should acknowledge that the data may overrepresent severe cases.
Q11: Data Extraction: Was there any inter-rater reliability check between the two independent researchers?
Q12: Figures: "Prima Flow-Chart" (Figure 1): Ensure alignment with PRISMA 2020 format.
Q13: Tables: "4yr+8/12" → Replace with "4 years 8 months" for clarity.
Q14: Odds Ratios: Extreme values (e.g., Phoenix Sepsis Score OR = 71.55) warrant verification for calculation errors.
Q15: Discussion: When discussing the use of protein-synthesis inhibitors, include potential mechanisms and whether data supports their routine use.
Q16: Discussion: "Steroid use was not statistically significant between groups." Discuss possible confounding factors that may influence this outcome.
Q17: Ethics: Please provide IRB approval code and date.
Author Response
Q1 :Abbreviations: Define iGAS in the list (currently missing).
R1: Thank you for your comment, we added it.
Q2: abstract: Only case report and case series were included." → Should be "Only case reports and case series were included."
R2: Thank you for your suggestion, we changed it.
Q3: abstract: "The identification of GAS from blood and Phoenix Sepsis Score ≥2 were statistically significantly associated..." → Should be "The identification of GAS from blood and a Phoenix Sepsis Score ≥2 were significantly associated..."
R3: Thank you for your advice, we changed it.
Q4: Case Reports: It will be good if the authors can describe more about signs of increased ICP, vital parameters, etc.
R4: Thank you for your suggestion, we added the vital parameters.
Q5: Case Reports: "4-year-8-month-old" → Hyphenation error; use "4 years and 8 months old.
R5: Thank you for your suggestion, we changed it.
Q6: Case 2: The decision-making process regarding therapy changes should be clarified. Why was vancomycin discontinued but linezolid added?
R6: Thank you for your suggestion. In case two, after the molecular identification of the pathogen we decided to stop vancomycin based on the local epidemiology of Streptococcus pyogenes. Linezolid was added as a protein synthesis inhibitor, considering that exotoxins are among the main virulence factors of this pathogen. While we acknowledge that there is currently no definitive evidence supporting the efficacy of Linezolid in this context, we decided to implement this therapeutic approach due to the patient's clinical deterioration and the severity of the case. We added these considerations in the section “discussion”.
Q7: Materials and Methods: Please explain what statistical software used for analysis. Also, please clarify whether logistic regression was considered for multivariable analysis.
R7: Thank you for your suggestion. We used Jamovi software, The statistical analysis has been improved with a multivariable logistic regression. The results are shown in the text.
Q8: Materials and Methods: please clarify the choice of a one-tailed t-test (rather than two-tailed) for comparing continuous variables.
R8: We chose a one-tailed t-test since our hypothesis was that the number of patients with lower age would be higher, CSF protein levels would be higher, CSF glucose levels lower, blood WBC count higher in the complicated course group compared to the uneventful course group.
Q9: Please address if the review protocol was registered in a database like PROSPERO.
R9: Thank you for your question, it wasn’t registered.
Q10: In the discussion, since only case reports and case series were included, you should acknowledge that the data may overrepresent severe cases.
R10: Thank you for your suggestion, we added it in the “limitation” section of the discussion.
Q11: Data Extraction: Was there any inter-rater reliability check between the two independent researchers?
R11: The rate of agreement among the two researchers was >80%; as reported in the text any discrepancies was discussed and resolved by consensus with all authors.
Q12: Figures: "Prima Flow-Chart" (Figure 1): Ensure alignment with PRISMA 2020 format.
R12: Thank you for your suggestion, we uploaded the integral PRISMA flow chart.
Q13: Tables: "4yr+8/12" → Replace with "4 years 8 months" for clarity.
R13: Thank you for your suggestion, we replaced it.
Q14: Odds Ratios: Extreme values (e.g., Phoenix Sepsis Score OR = 71.55) warrant verification for calculation errors.
R14: Thank you for your suggestion. We repeated the statistic analysis with JAMOVI software obtaining the same results. However, another reviewer suggested removing the comparison between survivors without sequels and lethal/sequels because of potential bias due to different follow-up periods among the analyzed papers thus that data was eliminated.
Q15: Discussion: When discussing the use of protein-synthesis inhibitors, include potential mechanisms and whether data supports their routine use.
R15: Thank you for your suggestion, we modified the paragraph.
Q16: Discussion: "Steroid use was not statistically significant between groups." Discuss possible confounding factors that may influence this outcome.
R16: Thank you for your suggestion, we revised the text.
Q17: Ethics: Please provide IRB approval code and date.
R17: Thank you for your suggestion, the manuscript was sent for revision to our Ethic Committee few weeks ago and we are awaiting for the approval for publication and related code. We will update the journal as soon as possible
Reviewer 2 Report
Comments and Suggestions for Authors
First, I wish to state that as a pediatric meningitis expert, the world does need more literature on GAS meningitis, so I am glad to see this. There are some things I really like in this paper, and some elements that need updating.
The formatting on Table 1 is not great-I would encourage you to align headers and in-text values.
Methods:
“Independent numerical variables that followed a normal distribution were compared between the two groups using a one-tailed t-test, with a significance level of 0.05.”
So here I do have questions. How did you know the variables followed a normal distribution? I see no evidence of Shapiro-wilk testing to support this. What did you do if it did not follow a normal distribution?
“This section may be divided by subheadings. It should provide a concise and precise The text continues here”
Remove this
Results
“Seven percent (4/57) of the cases had previous predisposing conditions for bacterial meningitis, such as medical devices, HIV infection, and prematurity [10,15,17,28].”
I would say Seven percent REPORTED predisposing conditions-your review cannot be sure such conditions did not exist and were not just unreported.
In general, I really question reporting sequelae. Consider reference 44, a reference you do not cite as evidence of sequelae. It is implied that this patient had no sequelae. However, the authors of this paper never say there was no sequelae, and even if they did, they would be time limited to the date of publication. I would remove sequelae, or possibly change it to something more observation by stating something like “XX number of cases noted sequelae”. This would include removing statistical comparisons of sequelae from Table 3.
Table 4: Please provide the raw numbers (how many people were in the age<1 group) for each value here. I would like to recalculate and check these values, as many readers would, but this is challenging without the raw numbers.
Discussion
“In those cases, the meningeal involvement could not be prevented even after adequate therapy of the primary infection, suggesting that more complex virulence factors may be implicated”
I disagree strongly. Consider that your data come from a review of literature. Which is more likely to be published-a severe case of meningitis, or a more general case of strep throat? I believe the effect seen here cannot be separated from publication bias. Remove this section.
I would move the sentence “However, in our analysis, steroid use was not statistically different between patients with a complete recovery and patient with sequelae[35,66-68]” to be immediately beneath “Dexamethasone may play a positive role in reducing brain edema and the inflammatory cascade. The results of its use in Streptococcus pneumoniae meningitis encourage its administration in GAS meningitis as well.” This will clarify some things. Some discussion of the fact that steroid treatment is debated in certain meningoencephalitis should be mentioned here with references.
“The limitations of our study include the small sample size, primarily due to the rarity of the disease, and the fact that the data were collected from different retrospective studies”
This needs to be dramatically expanded on. If something was not mentioned in a study, you often coded is as not happening. This is not necessarily true. You should, at a minimum, include a paragraph discussing this limitation.
Comments on the Quality of English LanguageThe English quality is not great, but passable.
Author Response
Q1: First, I wish to state that as a pediatric meningitis expert, the world does need more literature on GAS meningitis, so I am glad to see this. There are some things I really like in this paper, and some elements that need updating.
R1: Dear Reviewer we are glad to read your comments. Your suggestions are appreciated.
Q2 The formatting on Table 1 is not great-I would encourage you to align headers and in-text values.
R2:Thank you for your suggestion, we uploaded a new version of the table.
Q3 Methods: “Independent numerical variables that followed a normal distribution were compared between the two groups using a one-tailed t-test, with a significance level of 0.05.” So here I do have questions. How did you know the variables followed a normal distribution? I see no evidence of Shapiro-wilk testing to support this. What did you do if it did not follow a normal distribution?
R3: Thank you for your suggestion. We used Shapiro-Wilk test to check for the normal distribution of the variables that is automatically calculated by the software used (JAMOVI).
Q4:“This section may be divided by subheadings. It should provide a concise and precise The text continues here”.
R4: Thanks for noticing, we eliminated that part.
Q5: Results “Seven percent (4/57) of the cases had previous predisposing conditions for bacterial meningitis, such as medical devices, HIV infection, and prematurity [10,15,17,28].” I would say Seven percent REPORTED predisposing conditions-your review cannot be sure such conditions did not exist and were not just unreported. In general, I really question reporting sequelae. Consider reference 44, a reference you do not cite as evidence of sequelae. It is implied that this patient had no sequelae. However, the authors of this paper never say there was no sequelae, and even if they did, they would be time limited to the date of publication. I would remove sequelae, or possibly change it to something more observation by stating something like “XX number of cases noted sequelae”. This would include removing statistical comparisons of sequelae from Table 3.
R:5 Thank you for your suggestion. Regarding reference 44, the case report states: “The infant was discharged at age 2 months in good health; she was neurologically normal.” However, we agree with you that the main limitation of the study is that it described patients with different follow-up periods. We deleted the table as suggested.
Q6) Table 4: Please provide the raw numbers (how many people were in the age<1 group) for each value here. I would like to recalculate and check these values, as many readers would, but this is challenging without the raw numbers.
R6:The raw data of the variables analyzed in table 4 are reported in table 3. However, the complete dataset is available for consultation upon request.
Q7: Discussion. “In those cases, the meningeal involvement could not be prevented even after adequate therapy of the primary infection, suggesting that more complex virulence factors may be implicated”. I disagree strongly. Consider that your data come from a review of literature. Which is more likely to be published-a severe case of meningitis, or a more general case of strep throat? I believe the effect seen here cannot be separated from publication bias. Remove this section.
R7:Thank you for your observation, we agreed that the sentence could be confusing thus we removed it.
Q8: I would move the sentence “However, in our analysis, steroid use was not statistically different between patients with a complete recovery and patient with sequelae[35,66-68]” to be immediately beneath “Dexamethasone may play a positive role in reducing brain edema and the inflammatory cascade. The results of its use in Streptococcus pneumoniae meningitis encourage its administration in GAS meningitis as well.” This will clarify some things. Some discussion of the fact that steroid treatment is debated in certain meningoencephalitis should be mentioned here with references.
R8: Thank you for your suggestion, we modified the whole paragraph on the basis of the observations of all the reviewers.
Q9: “The limitations of our study include the small sample size, primarily due to the rarity of the disease, and the fact that the data were collected from different retrospective studies” This needs to be dramatically expanded on. If something was not mentioned in a study, you often coded is as not happening. This is not necessarily true. You should, at a minimum, include a paragraph discussing this limitation.
R9: Thank you for your suggestion, we expanded the comments on the limitations of the study.
Reviewer 3 Report
Comments and Suggestions for Authors
This manuscript focuses on group A streptococcal (GAS) meningitis in children, analysing its clinical features, complications, treatment and associated factors through case reports and systematic reviews to provide a reference for clinical management. The overall structure is complete and logical, but some improvements can be made.
- The discussion section mentions that the pathogenesis of GAS meningitis is not clear, but only briefly. It should be further combined with the progress of existing research, and the possible pathogenesis (e.g., how the virulence factor of GAS breaks through the blood-brain barrier and triggers the inflammatory response, etc.) should be explored in more depth, so as to enhance the theoretical depth of the discussion.
- Some of the charts and graphs have more content and the layout is dense, which affects the reading experience, and the font size in Figure 1 should be increased. The inclusion of appropriate cartoon illustrations is recommended.
- The discussion section should analyse the factors that may affect the future development, such as the mutation trend of GAS strains, the change of antibiotic resistance, and the adjustment of childhood vaccination procedures, etc., and the potential impact of these factors on the epidemiology, clinical manifestations, and therapeutic strategies of GAS meningitis.
Consideration may be given to having professional language editors or native English-speaking experts proofread the articles to improve the linguistic quality of the articles.
Author Response
This manuscript focuses on group A streptococcal (GAS) meningitis in children, analysing its clinical features, complications, treatment and associated factors through case reports and systematic reviews to provide a reference for clinical management. The overall structure is complete and logical, but some improvements can be made. Dear Reviewer, thank you for time and your advice.
Q1: The discussion section mentions that the pathogenesis of GAS meningitis is not clear, but only briefly. It should be further combined with the progress of existing research, and the possible pathogenesis (e.g., how the virulence factor of GAS breaks through the blood-brain barrier and triggers the inflammatory response, etc.) should be explored in more depth, so as to enhance the theoretical depth of the discussion.
R1: Thank you for your suggestion. We improved the discussion about potential mechanisms involved in GAS meningitis, although this topic is still debated in Literature.
Q2: Some of the charts and graphs have more content and the layout is dense, which affects the reading experience, and the font size in Figure 1 should be increased. The inclusion of appropriate cartoon illustrations is recommended.
R2: Thank you for your suggestion. We modified tables and figures to make them more appealing.
Q3: The discussion section should analyse the factors that may affect the future development, such as the mutation trend of GAS strains, the change of antibiotic resistance, and the adjustment of childhood vaccination procedures, etc., and the potential impact of these factors on the epidemiology, clinical manifestations, and therapeutic strategies of GAS meningitis.
R3: Thank you for your suggestion. We have revised this section with careful consideration of potential future issues.
Reviewer 4 Report
Comments and Suggestions for Authors
This work is interesting but it needs improvements:;
1: Species names MUST be in Italic
2: English revision throughout the entire manuscript, some mistakes and rephrasing
3: Table 3 and 4 in supplementary materials
4:Describe how bacteria has been identified and what instruments used and how antimicrobial pattern were had been determined
5: The description of case reports have to be inserted in results
Author Response
This work is interesting but it needs improvements:. Dear reviewer, thank you for your comment.
Q1: Species names MUST be in Italic.
R1 Thank you for your suggestion, we modified the text,
Q2: English revision throughout the entire manuscript, some mistakes and rephrasing
R2: Thank you for your suggestion, we revised it.
3Q: Table 3 and 4 in supplementary materials,
R3: Thank you for your suggestion. Table 3 and 4 were implemented based on the other reviewer’a suggestions. Table 4 was integrated with a logistic regression model.
Q4 :Describe how bacteria has been identified and what instruments used and how antimicrobial pattern were had been determined.
R4: Thank you for your advice.. The strain isolated in Case 1 was cultured on Columbia agar plate with 5% sheep blood (bioMérieux, Marcy-l'Étoile, France) and incubated at 37 °C overnight in 5% CO2 atmosphere. Colonies grown on agar plates were identified by using Matrix-Assisted Laser Desorption IonizationTime of Flight Mass Spectrometry (MALDI-TOF MS; Bruker Daltonics, Bremen, Germany). S. pyogenes strain was tested for antimicrobial susceptibility by VITEK®2 (bioMérieux, Marcyl'Étoile, France) automated system and results were interpreted according to clinical breakpoints based on the European Committee on Antimicrobial Susceptibility Testing (EUCAST) tables (version 13.0)1 . In Case 2, the text describes that the identification of the strain was done trough a Biofire Blood Culture Identification 2 (BCID2 ) Panel ((bioMérieux, Marcy l’Etoile, France). No growth was detected in the CSF and blood cultures, thus antimicrobial susceptibility test was not performed. If you agree, we could provide these descriptions as a supplementary file.
Q5: The description of case reports have to be inserted in results.
R5: Thank you for your suggestion. According to journal guidelines for authors, case reports were inserted before the materials and methods section.
Round 2
Reviewer 1 Report
Comments and Suggestions for Authors
accepted
Reviewer 3 Report
Comments and Suggestions for Authors
Overall, the authors responded positively to my comments, the manuscript has improved considerably, and it is recommended to be accepted.